# Study on Driver-Oriented Energy Management Strategy for Hybrid Heavy-Duty Off-Road Vehicles under Aggressive Transient Operating Condition

**Xu Wang** [1,2], **Ying Huang** [1,2,*] **and Jian Wang** [1]

1   School of Mechanical Engineering, Beijing Institute of Technology, Beijing 100081, China; wangxu_bit123@163.com (X.W.)
2   Beijing Institute of Technology Chongqing Innovation Center, Chongqing 401120, China
*   Correspondence: hy111@bit.edu.cn

**Abstract:** Hybrid heavy-duty off-road vehicles frequently experience rapid acceleration and deceleration, as well as frequent uphill and downhill motion. Consequently, the engine must withstand aggressive transients which may drastically worsen the fuel economy and even cause powertrain abnormal operation. When the engine cannot respond to the transient demand power quickly enough, the battery must compensate for the large amount of power shortage immediately, which may cause excessive battery current that adversely affects the battery safety and life span. In this paper, a nonlinear autoregressive with exogenous input neural network is used to recognize the driver's intention and translate it into subsequent vehicle speed. Combining energy management with vehicle speed control, a co-optimization-based driver-oriented energy management strategy for manned hybrid vehicles is proposed and applied to smooth the engine power to ensure efficient operation of the engine under severe transients and, at the same time, to regulate battery current to avoid overload. Simulation and the hardware-in-the-loop test demonstrate that, compared with the filter-based energy management strategy, the proposed strategy could yield a 38.7% decrease in engine transient variation and an 8.2% decrease in fuel consumption while avoiding battery overload. Compared with a sequential-optimization-based energy management strategy, which is recognized as a better strategy than a filter-based energy management strategy, the proposed strategy can achieve a 16.2% decrease in engine transient variation and a 3.2% decrease in fuel consumption.

**Keywords:** driver-oriented energy management; nonlinear model predictive control; driver's intention recognition; engine power smoothing; battery current regulation





## 1. Introduction

Automotive electrification and hybridization have shown promise in recent years, owing to low carbonization orientation and strict emission regulations [1]. Hybrid power systems have already been widely used in light-duty vehicles. It has also begun to be used in heavy-duty off-road vehicles along with the development of high-power traction motors and high-power-density batteries. However, these applications face some unique challenges. Due to the limitation of vehicle packaging space and weight, the battery capacity is limited, and the engine must bear most of the power demand. However, because of the poor dynamic response of diesel engines, excessive transient variation can make it difficult to control the engine to deliver the target power immediately, so the battery current may exceed the limit during power compensation. In some extreme cases, the engine may even stall because of overload [2]. Therefore, it is difficult for hybrid heavy-duty off-road vehicles to achieve the theoretical best fuel economy under aggressive transient conditions using conventional energy management strategy.

For hybrid electric vehicles, usually, powertrain-level energy management strategies including global optimization [3–5] and instantaneous optimization are used to achieve

the goals of both fuel consumption reduction and battery current regulation. Obviously, instantaneous optimization is more suitable for hybrid heavy-duty off-road vehicles. Instantaneous optimization, represented by the equivalent consumption minimization strategy (ECMS) [6,7], model predictive control (MPC) [8–11], and intelligent algorithms [12–14], has attracted much more attention recently because of its real-time application potential. Vehicle speed prediction is often used in instantaneous optimization, and usually, the predicted vehicle speed trajectory is used as the reference working condition for energy management optimization [15–18]. However, because energy management and vehicle speed control are decoupled, these speed prediction methods cannot accurately reflect the driver's intention on instantaneous vehicle speed. In addition, the dynamic characteristics of the engine are also not fully considered in optimization of energy management. As a result, they are not applicable to heavy-duty off-road vehicles, which normally take the diesel engine as its main power source and frequently experience aggressive transient operating conditions.

Among energy management strategies for heavy-duty off-road vehicles, Di Cairano et al. [2,19] emphasize the necessity of smoothing the engine power output to avoid aggressive engine transients and improve fuel economy. They add the engine power variation as a penalty item to the MPC objective function to smooth the engine operation. Compared with the adopted rule-based strategy, the engine operating points can be more closely distributed near the optimal brake specific fuel consumption (BSFC) line. Chen et al. [20] consider the dynamic characteristic of the engine when designing the MPC-based energy management strategy, which allows the engine power to change mildly. Kim et al. [21–23] develop a filter-based energy management strategy for a series diesel–electric hybrid military vehicle, in which a first-order low-pass filter is used to make the engine account for the low-frequency portion of the power and the battery pack for the high-frequency portion. Therefore, aggressive engine transients were significantly reduced. However, the phase delay introduced by the low-pass filter may result in sluggish engine response. The above studies all focus on smoothing engine power demand but ignore the resulting adverse effects on the battery, i.e., the battery current may be too high or even over the limit. To the authors' knowledge, few studies exist regarding smoothing the engine power and regulating battery current simultaneously in the control of hybrid heavy-duty off-road vehicles.

The emerging technologies of connectivity and automatic driving make it possible to plan vehicle speed and optimize energy management strategy simultaneously [24], this is so-called vehicle-level energy management. There are two main types of vehicle-level energy management: sequential optimization and co-optimization. Sequential optimization is the most applied method, during which the two issues are decoupled into two layers. In the first layer, according to the planned vehicle speed, the demand power is regulated and transmitted to the energy management system as a target, and in the second layer, it is distributed to different power sources [25–28]. However, these two issues are addressed together in co-optimization. Kim et al. [29] conducted a co-optimization of vehicle speed planning and energy management for a fuel cell–battery hybrid military vehicle and compared the co-optimization results with that of the sequential optimization. The result indicates that co-optimization has a higher energy-saving potential than sequential optimization. However, speed planning cannot be implemented for manned vehicles. How to apply vehicle-level energy management to manned vehicles is still an open issue.

Energy management of hybrid electric vehicles embodies the concept of sustainability, specifically in the following aspects: (1) High energy efficiency: A hybrid electric vehicle adopts dual power systems, namely engine and battery, and controls the switching and cooperation of the two power systems in real time through the energy management system. Compared with traditional vehicles, hybrid vehicles can use fuel energy more efficiently, which greatly improves the energy utilization efficiency. (2) Reduce emissions: Hybrid electric vehicles can monitor and optimize the power system in real time in energy management, which greatly reduces vehicle emissions. In the pure electric mode, the zero-emission

capability of the automobile is further exerted. This provides a certain guarantee for environmental protection. (3) Improve service life: Through energy management, the hybrid electric vehicle prolongs the battery service life. This saving behavior not only reduces waste but also protects the environment. Therefore, the energy management of hybrid electric vehicles fully demonstrates the consideration and embodiment of sustainability by improving energy utilization efficiency, reducing emissions, and prolonging battery service life.

Motivated by vehicle-level energy management, and combining energy management with vehicle speed control, a vehicle-level driver-oriented energy management strategy is proposed here to smooth engine power variation and avoid battery current overload simultaneously. Specific ideas and contributions include:

1. A nonlinear autoregressive with exogenous input (NARX) neural network is trained to recognize the driver's driving intention and translate it into the expected subsequent vehicle speed trajectory, which will be used as the tracking reference of the MPC controller.
2. A nonlinear MPC controller is designed to track the reference vehicle speed and realize energy management. By using MPC, the driving force and engine power are co-optimized, which allows engine power to follow demand power well so that battery current overload is avoided. The penalty weights are used to coordinate the variations in driving force and engine power, which leads to improvement of engine power smoothing and further decrease in transient battery current. The weights in the objective function are adaptively selected according to the vehicle working condition classification to ensure that the demand power variation can match the engine dynamic characteristic under different transient conditions.
3. The effectiveness of the proposed strategy is verified through simulation and the hardware-in-the-loop (HiL) test in an actual off-road driving cycle with drastic power fluctuations.

The rest of this article is organized as follows. In Section 2, simulation-oriented dynamic vehicle models are introduced. In Section 3, a co-optimization-based driver-oriented energy management strategy is introduced. Section 4 presents analysis and comparison of the simulation results and HiL test results. Finally, the conclusions are presented in Section 5.

## 2. Simulation-Oriented Vehicle Model

The object of this paper is a series hybrid heavy-duty off-road vehicle, with the configuration shown in Figure 1. This vehicle has a total weight of 30,000 kg and is designed for harsh terrain and environment. The front power chain includes a primary power source, composed of a diesel engine and a generator; an auxiliary power source, composed of a battery; and a gear box located between the engine and the generator to match the operating ranges of the two components. The rear power chain primarily consists of two driving motors and two dual-speed transmissions. The main vehicle parameters are shown in Table 1. In Sections 2.1–2.4, a powertrain dynamic model suitable for transient study will be introduced and described in detail.

**Table 1.** Main parameters of the vehicle.

| Parameter | Description |
|---|---|
| Curb weight | 30,000 kg |
| Rolling resistance coefficient | 0.1 |
| Frontal area | 4 m$^2$ |
| Drag coefficient | 1 |
| Rotating mass conversion factor | 1.2 |
| Length × width × height | 7.5 m × 3 m × 2 m |
| Engine | 8 V diesel engine; normal power: 550 kW |

**Table 1.** *Cont.*

| Parameter | Description |
| --- | --- |
| Gear box | Ratio: 1.9 |
| Generator | Permanent magnet synchronous motor; normal power: 500 kW; peak power: 600 kW |
| Motor | Permanent magnet synchronous motor; normal power: 200 kW; peak power: 313 kW |
| Battery | Lithium iron phosphate; layout: 2P246S; capacity: 96 Ah; voltage: 900 V |
| Transmission | Dual-speed mechanical transmission; ratios: 28.78, 12.02 |

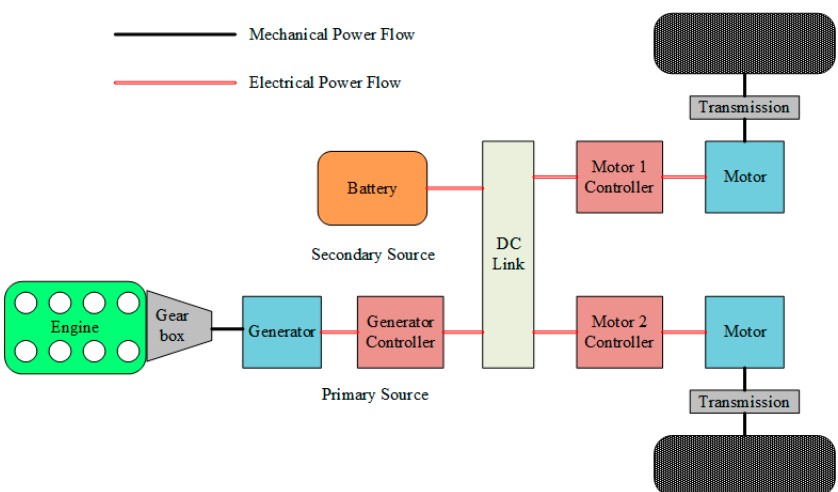

**Figure 1.** Powertrain configuration.

### 2.1. Engine Model

To simulate the engine characteristics under aggressive transients with relatively less computational burden, we adopt a mean value model, which is built and calibrated based on our previous work [30]. The structure of the engine model is shown in Figure 2. The engine model consists of six parts. The mass, the pressure, and the temperature of the air flowing into the cylinder are calculated by the compressor model, intercooler model, and intake manifold model, respectively, and the injected fuel mass is calculated by the controller model. Then, the combustion model converts heat into mechanical energy to drive the crankshaft. The exhaust gas enters the turbine model after passing through the exhaust manifold model to recover energy, and the torque converted from the exhaust gas energy drives the compressor rotation, thus realizing the closed loop of the whole simulation.

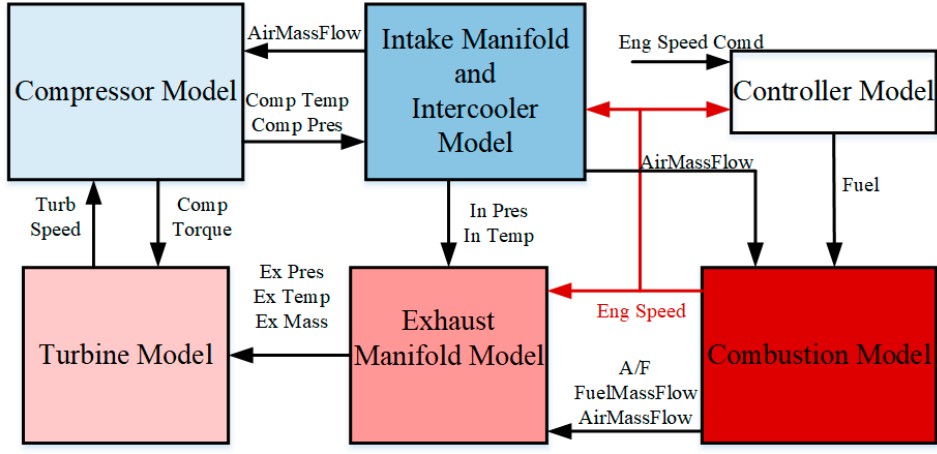

**Figure 2.** Engine model.

### 2.1.1. Turbocharger Model

The turbocharger is composed of a compressor, a turbine, and a rotor. The efficiency and pressure ratio of the compressor are obtained from experimental data. The compressor outlet temperature and the torque required to drive the compressor are calculated through the isentropic thermal efficiency relationships. These relationships are presented in Equations (1)–(4) [31]:

$$\pi_c = f\left(\dot{m}_c, N_{tc}\right) \tag{1}$$

$$\eta_c = f\left(\dot{m}_c, N_{tc}\right) \tag{2}$$

$$T_{c,out} = T_{c,in}\left(1 + \frac{1}{\eta_c}\left(\pi_c^{\frac{k-1}{k}} - 1\right)\right) \tag{3}$$

$$M_c = \frac{\dot{m}_c C_{pa} T_{c,in}}{\eta_c \omega_{tc}}\left(\pi_c^{\frac{k-1}{k}} - 1\right) \tag{4}$$

where $\pi_c$ represents the pressure ratio, $\eta_c$ is the compressor efficiency, $\dot{m}_c$ is the air mass flow rate (kg/s), $N_{tc}$ is the rotor speed (n/min), $\omega_{tc}$ is the rotor speed (rad/s), $T_{c,in}$ and $T_{c,out}$ are the compressor inlet and outlet temperatures (K), $k$ is the adiabatic exponent, $M_c$ is the compressor torque (N·m), and $C_{pa}$ is the specific heat of air at constant pressure (kJ/(kg·K)), which was assumed to be constant.

The turbine modeling principle was similar to that of the compressor, and could be expressed by Equations (5)–(7) [31]:

$$M_t = \frac{\dot{m}_t C_{pe} T_{t,in}}{1000 \eta_t \omega_{tc}}\left(\pi_t^{\frac{k-1}{k}} - 1\right) \tag{5}$$

$$M_t = \frac{\dot{m}_t C_{pe} T_{t,in}}{1000 \eta_t \omega_{tc}}\left(\pi_t^{\frac{k-1}{k}} - 1\right) \tag{6}$$

$$\dot{m}_t = \dot{m}_a + \dot{m}_f \tag{7}$$

where $M_t$ represents the turbine torque (N·m); $\dot{m}_t$, $\dot{m}_a$, and $\dot{m}_f$ are the exhaust gas, air, and fuel mass flow rates (g/s), respectively; and $C_{pe}$ is the specific heat of the exhaust gas at constant pressure (kJ/(kg·K)), which was assumed to be constant. $T_{t,in}$ is the turbine inlet temperature (K).

The rotor dynamic model was obtained:

$$M_t - M_c = J_{tc}\dot{\omega}_{tc} \tag{8}$$

where $J_{tc}$ is the inertia of the turbocharger rotor.

### 2.1.2. Intercooler Model

The intercooler model primarily simulates the changes in the gas temperature and pressure. The temperature relationship could be expressed by Equations (9) and (10) [32]:

$$T_{im} = T_{c,out}(1 - \eta_{cool}) + \eta_{cool} T_{coolant} \tag{9}$$

$$\eta_{cool} = f\left(N_e, \dot{m}_a\right) \tag{10}$$

where $T_{im}$ represents the intake manifold temperature (K); $T_{coolant}$ is the coolant temperature, which was approximately 433 K; and $\eta_{cool}$ is the intercooler efficiency, which could be expressed in tabular form alongside the engine speed and the air mass flow rate.

The gas experiences a pressure drop after passing through the intercooler because flow resistance is present. The pressure drop is calculated using Equation (11) [31]:

$$\Delta p = \frac{\varepsilon \dot{m}_c^2}{\rho_{im}} = \frac{\varepsilon \dot{m}_c^2 R T_{im}}{p_{im}} \tag{11}$$

where $\varepsilon$ is the friction factor of the intercooler, $R$ is the ideal gas constant, $\rho_{im}$ is the intake mass density (kg/m$^3$), and $p_{im}$ is the intake manifold pressure (Pa).

### 2.1.3. Intake and Exhaust Manifold Model

Based on the first law of thermodynamics and the ideal gas state equation, the relationship between the pressure and the air flow rate in the intake manifold is obtained, as shown in Equations (12)–(14) [31]:

$$\dot{p}_{im} = \frac{T_{im}(\dot{m}_c - \dot{m}_a)kR}{V_{im}} \tag{12}$$

$$\dot{m}_a = \frac{V_d}{120RT_{im}}\eta_v p_{im} N_e \tag{13}$$

$$\eta_v = f(N_e, p_{im}) \tag{14}$$

where $V_{im}$ and $V_d$ are the intake manifold and cylinder volumes (m$^3$), respectively, and $\eta_v$ is the volumetric efficiency, which could be expressed in tabular form alongside the engine speed and the intake manifold pressure.

The pressure of the exhaust manifold is obtained using an empirical formula [30], as shown in Equation (15):

$$p_{em} = \frac{1}{2}\left(p_{im} + \sqrt{p_{im}^2 - 4\dot{m}_{em}^2\beta(2T_{im} + \Delta T_{em})}\right) \tag{15}$$

where $\Delta T_{em}$ represents the temperature rise (K), and $\beta$ is an empirical parameter.

### 2.1.4. Cylinder Model

The cylinder model is primarily used to calculate the indicated torque and the engine speed. The indicated torque is calculated using Equations (16) and (17) [31].

$$M_i = \frac{9.55\left(\dot{m}_f \cdot Hu \cdot \eta_i\right)}{N_e} \tag{16}$$

$$\eta_i = f\left(N_e, \frac{A}{F}\right) \tag{17}$$

Engine speed is obtained from Newton's second law:

$$M_i - M_f - M_l = J\ddot{\varphi} \tag{18}$$

$$M_f = f(N_e, M_l) \tag{19}$$

where $M_f$ represents the friction torque (N·m), which is obtained from a table using the engine speed ($N_e$) and the load torque ($M_l$); $J$ is the inertia of the rotating parts (kg·m$^2$); and $\varphi$ is the crankshaft angle (rad).

### 2.1.5. Engine Model Validation

The mean value engine model is built based on our previous work [31], and each part of the model is calibrated with test data so that the error could be within 5%. As the focus of this paper is not on calibrating the model, only the results are shown here. As shown in Figure 3, the engine model had high accuracy.

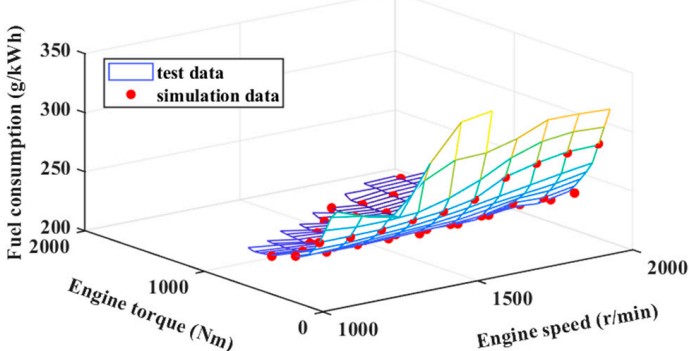

**Figure 3.** Engine model validation.

### 2.2. Generator and Driving Motor Model

The generator and the driving motors were permanent magnet synchronous motors (PMSMs); the generator is a surface-mounted PMSM, and the driving motor is an interior PMSM. A *dq* axis model is used to accurately simulate the motor performance.

To simplify the calculations, the variables in a three-phase coordinate system (the *abc* coordinate system) were transformed into a two-phase rotating coordinate system (the *dq* coordinate system) using Clarke and Park transforms. The motor voltage, flux linkage, electromagnetic torque, and kinematic equations in the *dq* coordinate system were used for modeling, as shown in Equations (20)–(25) [32]. Then, the variables in the *dq* coordinate system were transformed back to the *abc* coordinate system using Park and Clarke inverse transforms. Figure 4 presents a diagram of the motor model.

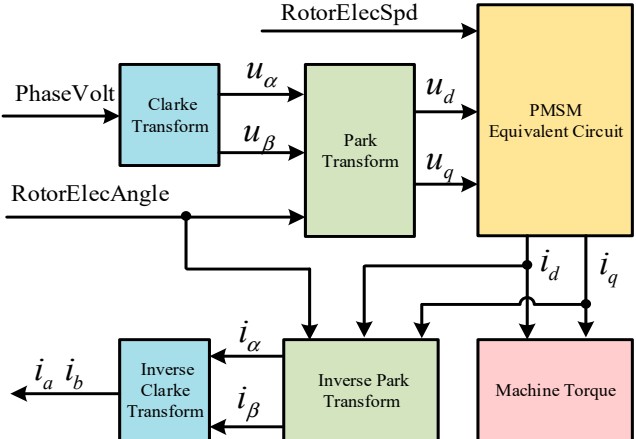

**Figure 4.** Motor model.

According to the principle of holding power constant throughout a transform, the Clarke and Park transforms could be expressed by Equations (20)–(22):

$$\begin{bmatrix} i_d \\ i_q \end{bmatrix} = C_{2s \to 2r} \ C_{3s \to 2s} \begin{bmatrix} i_a \\ i_b \\ i_c \end{bmatrix} \qquad (20)$$

$$C_{3s \to 2s} = \sqrt{\frac{2}{3}} \begin{bmatrix} 1 & -\frac{1}{2} & -\frac{1}{2} \\ 0 & \frac{\sqrt{3}}{2} & -\frac{\sqrt{3}}{2} \end{bmatrix} \tag{21}$$

$$C_{2s \to 2r} = \begin{bmatrix} \cos\theta & \sin\theta \\ -\sin\theta & \cos\theta \end{bmatrix} \tag{22}$$

where $C_{3s \to 2s}$ is the Clarke transform matrix; $C_{2s \to 2r}$ is the Park transform matrix; $i_a$, $i_b$, and $i_c$ represent the stator $a$, $b$, and $c$ phase currents (A), respectively; and $i_d$ and $i_q$ are the $d$ and $q$ axis currents (A), respectively.

Equation (23) provides the stator voltage equations:

$$\begin{cases} u_d = Ri_d + \frac{d\psi_d}{dt} - \psi_q \omega_{ele} \\ u_q = Ri_q + \frac{d\psi_q}{dt} + \psi_d \omega_{ele} \end{cases} \tag{23}$$

where $u_d$ and $u_q$ are the $d$ and $q$ axis voltages (V), respectively; $R$ is the stator resistance ($\Omega$); $\psi_d$ and $\psi_q$ are the $d$ and $q$ axis stator flux linkages (Wb), respectively; and $\omega_{ele}$ is the rotor electric angular velocity (rad/s).

Equation (24) expresses the electromagnetic torque:

$$T_{ele} = \frac{3}{2} p_n \left( \psi_d i_q - \psi_q i_d \right) \tag{24}$$

where $T_{ele}$ is the electromagnetic torque (Nm), and $p_n$ is the number of pole pairs.

The kinematic equation is shown in Equation (25):

$$T_{ele} - T_l - B\omega_r = J\frac{d\omega_r}{dt} \tag{25}$$

where $T_l$ is the load torque (Nm), $B$ is the resistance coefficient, and $\omega_r$ is the mechanical angular velocity (rad/s).

The field-oriented control is adopted for the motors [33]. When the target torque of the driving motor is negative, it is the regenerative braking mode. The target torque of the driving motor is calculated by the upper controller and is limited by the maximum torque of the motor at the current state and the torque calculated from the maximum battery charging power.

### 2.3. Battery Model

The open circuit voltage, $U_{oc}$, and the internal resistance, $R_b$, were the primary considerations in the battery model, as shown in Figure 5.

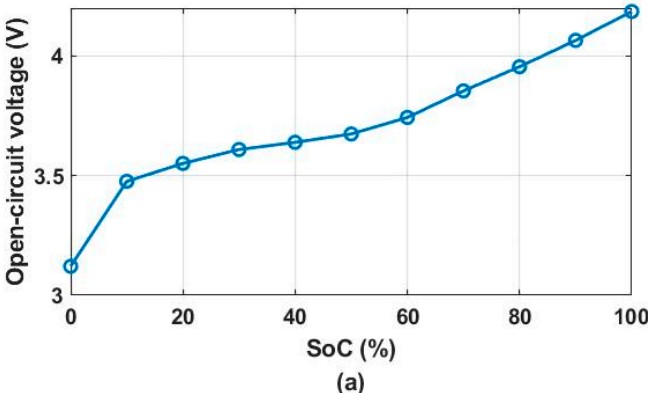
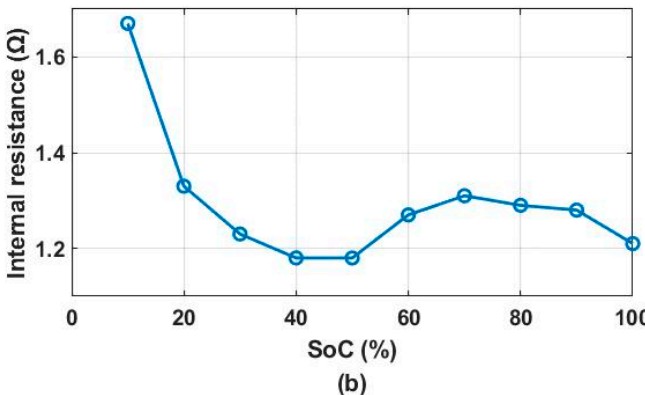

**Figure 5.** Battery characteristics: (**a**) open-circuit voltage; (**b**) internal resistance.

The model can be described using Equations (26)–(28):

$$I_b = -\frac{U_{oc} - \sqrt{U_{oc}^2 - 4P_b R_b}}{2R_b} \tag{26}$$

$$SoC = SoC_0 - \int \frac{I_b}{Q} dt \tag{27}$$

$$\dot{SoC} = -\frac{U_{oc} - \sqrt{U_{oc}^2 - 4P_b R_b}}{2R_b Q} \tag{28}$$

where $I_b$ and $P_b$ are the battery current (A) and battery power (kW) (positive for discharge and negative for charge), and $Q$ is the battery nominal capacity (Ah).

### 2.4. Vehicle Longitudinal Dynamics Model

The vehicle longitudinal dynamics could be described by the differential equation in Equation (29):

$$\delta m \frac{dv}{dt} = F_t - mgf\cos\alpha - mg\sin\alpha - \frac{1}{2}\rho C_D A v^2 \tag{29}$$

where $m$ represents the curb weight (kg); $v$ is the vehicle speed (m/s); $f$ is the rolling resistance coefficient; $\alpha$ is the road slope (°); $\rho$ is the air density (kg/m$^3$), which is assumed to be constant; $C_D$ is the drag coefficient; and $A$ denotes the frontal area (m$^2$).

### 2.5. Control Problem Analysis

To show the control problem more concretely, the vehicle model developed in Section 2 is used in simulations. The powertrain-level filter-based energy management strategy [22,23] is adopted as the benchmark in which first-order low-pass filtering is used to make the engine account for the low-frequency portion of the power, and the battery pack is responsible for the high-frequency portion.

#### 2.5.1. Driving Cycle Used for Simulation

There is no standard test cycle for heavy-duty off-road vehicles, so the vehicle speed and slope obtained in real tests are used as simulation conditions. As shown in Figure 6, a portion of the off-road vehicle's driving cycle with road slope information is used to simulate the filter-based control strategies. The total distance, which is 10.76 km, is covered in 1087 s.

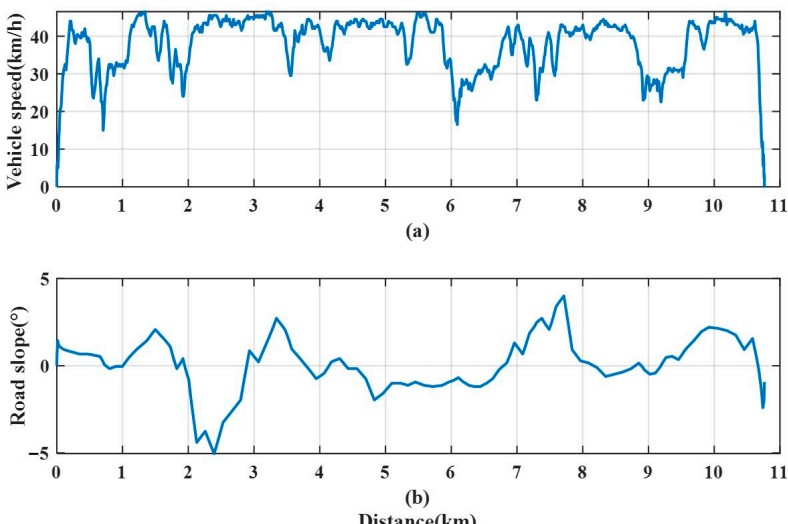

**Figure 6.** Driving cycle used for simulation: (**a**) vehicle speed; (**b**) road slope.

### 2.5.2. Results of Filter-Based Energy Management

Figure 7 shows the vehicle performance when the filter-based energy management strategy is used. The condition indicated by circle 1 is the acceleration when going uphill. The demand power rises rapidly, but the engine power adjusts slowly, so the insufficient power is supplemented by the battery, causing the battery discharge current to exceed the limit. The condition indicated by circle 2 is the deceleration when going downhill. Affected by the phase delay of the low-pass filter, the engine power fails to reduce in time, causing the battery charging current to exceed the limit. Undoubtedly, excessive battery current has a negative impact on battery life and safety.

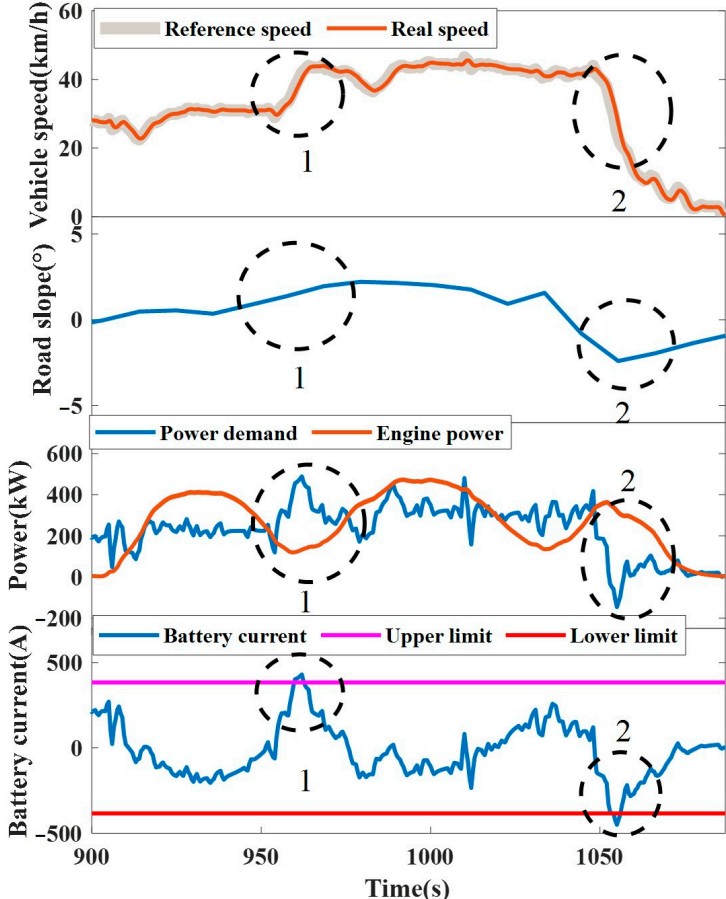

**Figure 7.** Problems with the filter-based strategy.

## 3. Co-Optimization-Based Driver-Oriented Energy Management Strategy

Figure 8 presents the control scheme of the co-optimization-based driver-oriented energy management strategy. The driver's intention is recognized using a NARX neural network. A nonlinear MPC controller is adopted to achieve co-optimization of demanded power and engine-generator power through specially designed objective function. In addition to the common goals of vehicle speed tracking, SoC maintenance, and fuel consumption reduction, matching the demand power variations with the dynamic characteristic of the engine is also considered as an additional goal. So, the driving force variation ($\Delta Fm$) and engine power variation ($\Delta Pg$) are added to the objective function as penalties. Working condition classification is carried out and reasonable weights are calibrated according to the characteristics of different working conditions. By assigning reasonable weights in the objective function under different vehicle working conditions, the demanded power variations are modified so that the aggressive engine transient operation can be avoided. At the same time, excessive battery current can also be avoided.

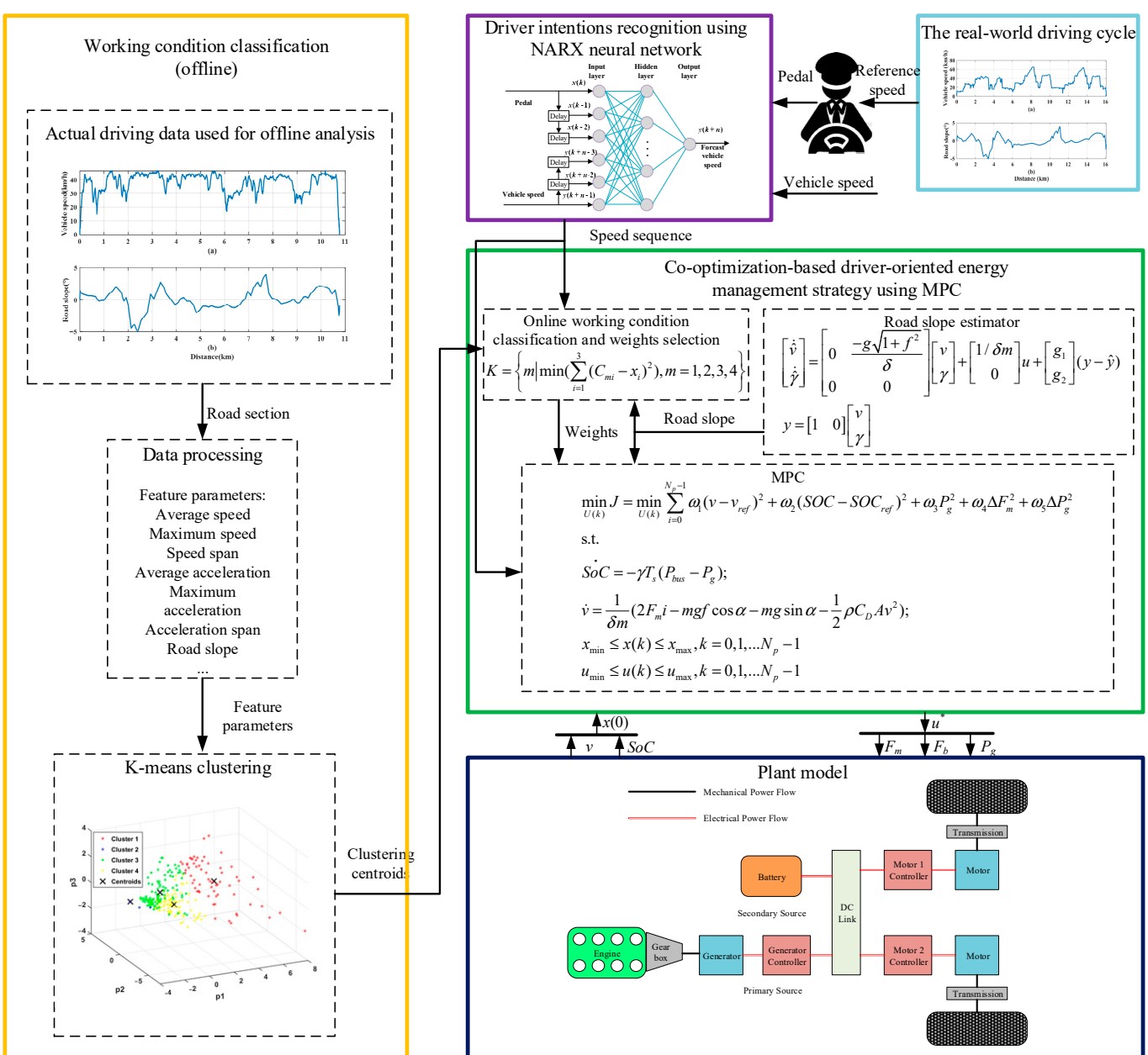

**Figure 8.** Scheme of co-optimization-based driver-oriented energy management strategy.

### 3.1. Driving Intention Recognition Using a NARX Neural Network

A NARX neural network, which includes tapped delay lines (TDLs), is used to translate pedal signals into a sequence of desired future vehicle speed. The past input and output states were recorded by TDL units. Combined with the nonlinear mapping characteristics of multi-layer perception, the state value of the next moment could be predicted from the past states:

$$y(t) = f(y(t-1), y(t-2), \cdots, y(t-n_y), u(t-1), u(t-2), \cdots, u(t-n_u)) \quad (30)$$

where $f(\cdot)$ represents a nonlinear function, $y(t - n_y)$ represents the neural network output and its delay $n_y$, and $u(t - n_u)$ represents the neural network input and its delay $n_u$.

The neural network predicted the subsequent vehicle speed using the past pedal signal and vehicle speed. To improve the prediction accuracy, an open-loop structure is adopted for the NARX neural network, which replaced the output feedback with the actual vehicle speed. Multi-step predictions are achieved by connecting neural networks in series.

The neural network's structure is shown in Figure 9. Increasing the number of hidden layer nodes and delays is beneficial for improving the training accuracy, but the complex neural network structure is prone to over-fitting. Considering the prediction accuracy and generalization comprehensively, it is determined in this study that there would be 10 hidden layer nodes and three delays, that is, the acceleration and vehicle speed from the past three seconds are used to predict the subsequent vehicle speed.

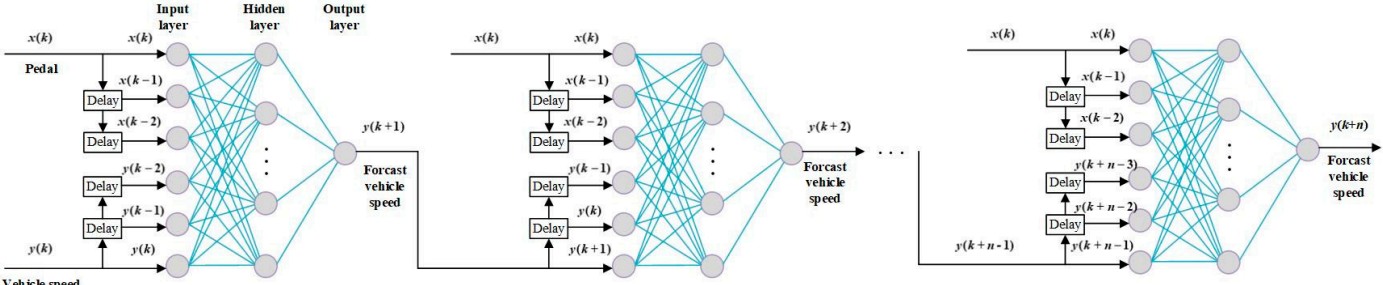

**Figure 9.** Neural network for vehicle speed prediction.

A real vehicle test data set, with a total length of 6340 s, is chosen to train the NARX neural network, as shown in Figure 10. Vehicle speeds are predicted for the next 10 s. Figure 11 shows the 10-step vehicle speed prediction errors. The error for the tenth step is the largest, but it is still within $\pm 5$ km/h, indicating that the prediction accuracy was acceptable.

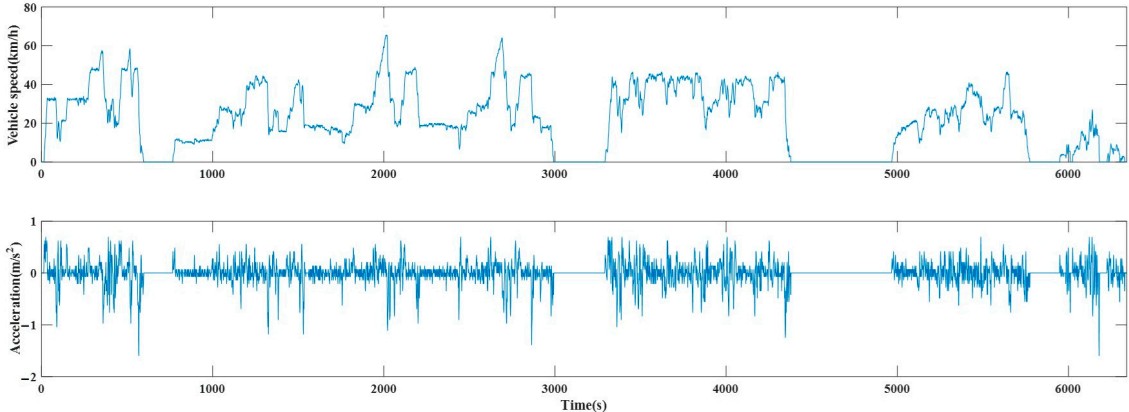

**Figure 10.** Training data.

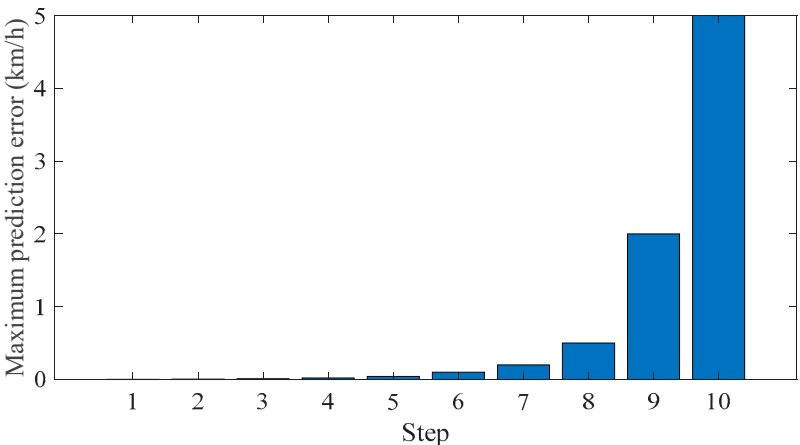

**Figure 11.** Maximum vehicle speed prediction error for different steps.

### 3.2. Road Slope Estimation Based on Luenberger Observer

In order to reduce the error of the prediction model used in MPC and improve the robustness of the controller, it is important to obtain the road slope in real time. In this study, a Luenberger observer is selected to instantaneously estimate the road slope.

The expression of the vehicle longitudinal dynamics in Equation (29) is mathematically changed, as shown in Equation (31).

$$\delta m \frac{dv}{dt} = F_t - F_w - mg\sqrt{1+f^2}sin(\alpha + \beta) \tag{31}$$

where $mg\sqrt{1+f^2}sin(\alpha + \beta) = mgf\cos\alpha + mg\sin\alpha$, $f = \tan\beta$, $F_w = \frac{1}{2}\rho C_D Av^2$.

Let $\gamma = \sin(\alpha + \beta)$, $u = F_t - F_w$; the state space equation can be obtained:

$$\begin{cases} \begin{bmatrix} \dot{\hat{v}} \\ \dot{\hat{\gamma}} \end{bmatrix} = \begin{bmatrix} 0 & \frac{-g\sqrt{1+f^2}}{\delta} \\ 0 & 0 \end{bmatrix} \begin{bmatrix} v \\ \gamma \end{bmatrix} + \begin{bmatrix} 1/\delta m \\ 0 \end{bmatrix} u + \begin{bmatrix} g_1 \\ g_2 \end{bmatrix}(y - \hat{y}) \\ y = \begin{bmatrix} 1 & 0 \end{bmatrix} \begin{bmatrix} v \\ \gamma \end{bmatrix} \end{cases} \tag{32}$$

$g_1$ and $g_2$ are determined using the pole assignment method, and the relationships between $g_1$ and $g_2$ and the eigenvalues $\lambda_1$ and $\lambda_2$ can be expressed by Equation (33):

$$g_1 = -(\lambda_1 + \lambda_2), g_2 = -\frac{\lambda_1\lambda_2\delta}{g\sqrt{1+f^2}} \tag{33}$$

To obtain a stable observer system, $\lambda_1$ and $\lambda_2$ should be negative, and their specific values can be calibrated according to the estimation accuracy requirements. Figure 12 presents a comparison diagram of road slope estimations with different eigenvalues. The farther the eigenvalues are from the imaginary axis, the faster the system responds. However, the system will also be more sensitive to noise. Considering these factors, $\lambda_1 = -1.2$ and $\lambda_2 = -1.5$ are selected.

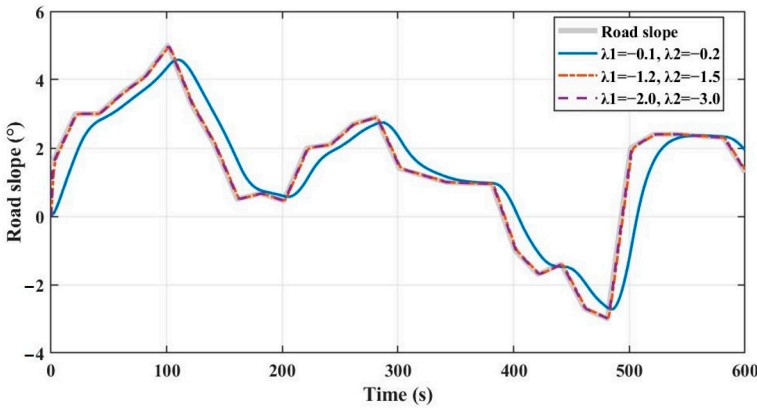

**Figure 12.** Road slope estimation.

### 3.3. Working Condition Classification

In order to set reasonable weights of the objective function under different transient conditions, working condition classification is carried out, which is introduced below.

The 6340 s of vehicle speed data and their corresponding road slope data presented in Figure 13 are divided into 634 segments at intervals of 10 s, and the feature parameters of each segment are extracted. Selected feature parameters are shown in Table 2, and the feature parameters obtained are 10-dimensional. To reduce the number of computations during the clustering process, a principal component analysis (PCA) is used to reduce the number of dimensions in the data. Three feature parameters with a cumulative contribution rate of more than 80% are selected as the basic data for the clustering analysis. K-means

clustering is used to classify the working conditions characterized by drop-dimensional feature parameters, and clustering centroids are saved for the online calculation process. The classification results are shown in Figure 14 and the characteristics of various working conditions are summarized in Table 3.

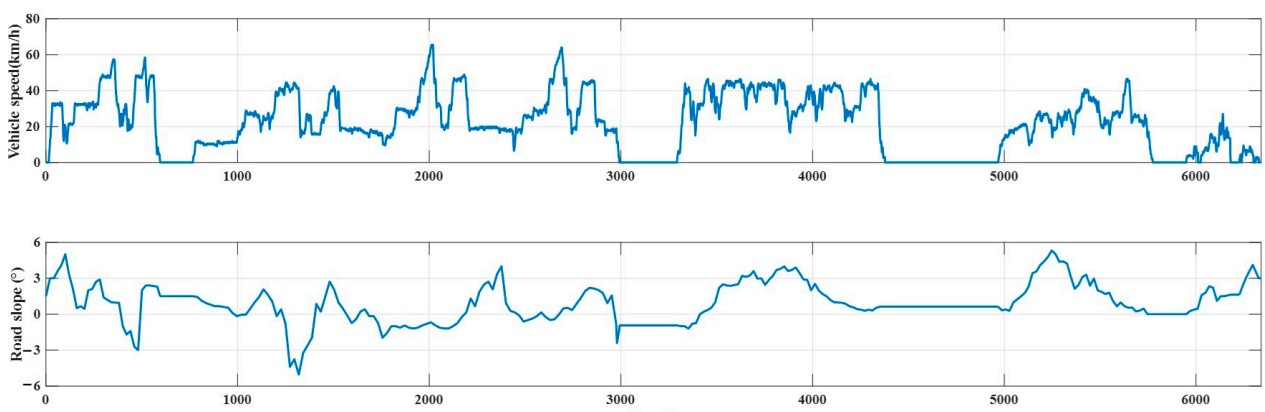

**Figure 13.** Working condition for offline analysis.

**Table 2.** Feature parameters.

| Number | Feature Parameter |
|---|---|
| 1 | Average speed |
| 2 | Maximum speed |
| 3 | Speed span |
| 4 | Average acceleration |
| 5 | Maximum acceleration |
| 6 | Accelerated proportion |
| 7 | Average deceleration |
| 8 | Maximum deceleration |
| 9 | Deceleration proportion |
| 10 | Road slope |

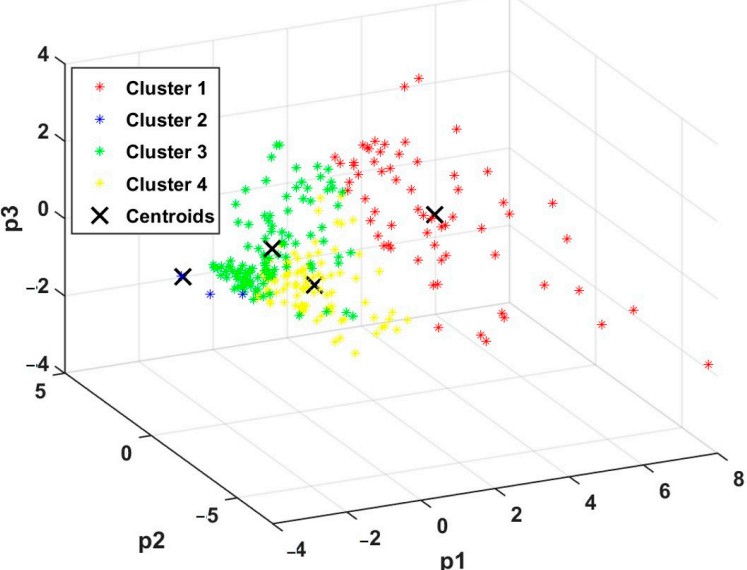

**Figure 14.** Cluster assignments and centroids.

**Table 3.** Classification of working condition.

| Working Condition Feature | Example |
|---|---|
| 1<br>Low power demand<br>(0 < P < 143 kW) |  |
| 2<br>Parking<br>(P = 0) |  |
| 3<br>High power demand<br>(435 kW < P < 550 kW) |  |
| 4<br>Medium power demand<br>(143 kW < P < 435 kW) |  |

### 3.4. MPC Design

The vehicle speed prediction model is built based on the vehicle longitudinal dynamics equation, as shown in Equation (34):

$$\dot{v} = \frac{1}{\delta m}\left(2F_m i - mgf\cos\alpha - mg\sin\alpha - \frac{1}{2}\rho C_D A v^2\right) \tag{34}$$

where $F_m$ is the driving motor force, and $i$ is the transmission ratio.

Usually, an equivalent circuit model, as shown in Equation (28), is adopted for SoC predictions [34]. However, this complex nonlinear model greatly increased the computation complexity, so a simplified linear regression (LR) model is used, as shown in Equation (35):

$$\dot{SoC} = -\gamma T_s P_b = -\gamma T_s \left(P_{bus} - P_g\right) \tag{35}$$

where $T_s$ represents the sampling time, and $P_b$, $P_g$, and $P_{bus}$ are the battery power, generator power, and DC bus power, respectively.

Figure 15 presents a comparison between the Rint model and the LR model during 10 s of operation. The maximum SoC prediction error for the LR model did not exceed

0.05% within the 10 s. With this model simplification, the accuracy requirement is met in the prediction horizon.

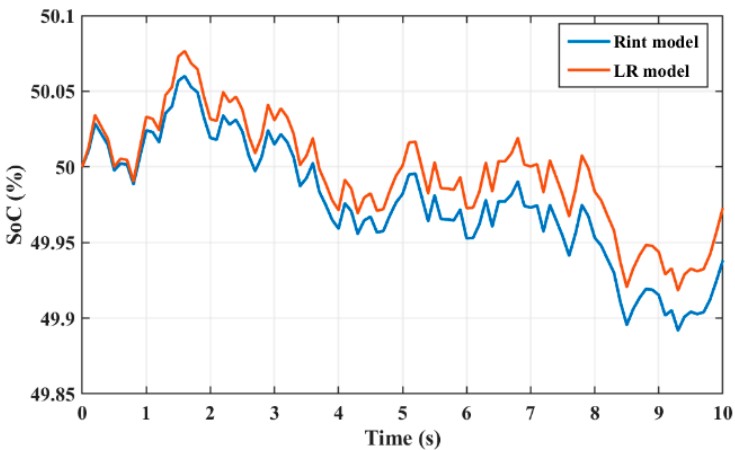

**Figure 15.** Comparison between the Rint model and the LR model.

Equations (34) and (35) constitute the predictive model required for co-optimization, and the state variables are $x = [v, SoC]^{\mathrm{T}}$, the control inputs were $u = [F_m, P_g]^{\mathrm{T}}$, and the measurable disturbance is $d = \alpha$.

All problems of driver-oriented energy management are comprehensively considered during the co-optimization, and the optimization problem can be expressed by Equation (36).

$$\min_{U(k)} J = \min_{U(k)} \sum_{i=0}^{N_p-1} \omega_1 \left(v - v_{ref}\right)^2 + \omega_2 \left(SoC - SoC_{ref}\right)^2 + \omega_3 \dot{m}_f^2 + \omega_4 \Delta F_m^2 + \omega_5 \Delta P_g^2$$

s.t.

$$x_{\min} \le x(k) \le x_{\max}, k = 0, 1, \dots N_p - 1$$
$$u_{\min} \le u(k) \le u_{\max}, k = 0, 1, \dots N_p - 1$$

(36)

Because the BSFC tracking strategy is adopted, the relationship between the engine fuel mass flow rate, $\dot{m}_f$ (g/s), and the generator power, $P_g$ (kW), is approximately linear. This is especially true for a medium load, as shown in Figure 16.

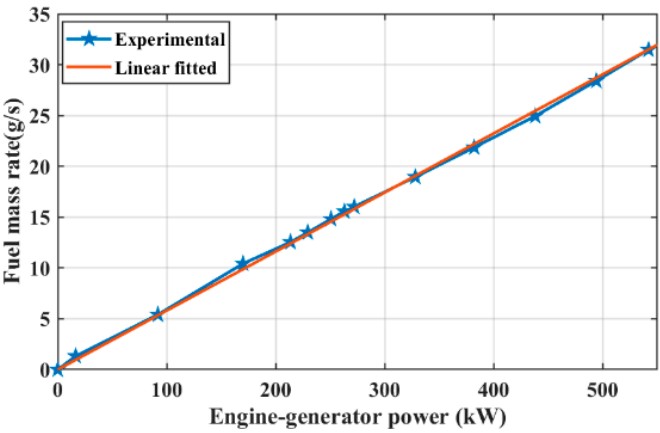

**Figure 16.** Fuel mass rate.

The fuel mass flow rate can be expressed by Equation (37):

$$\dot{m}_f = k_r P_g$$

(37)

where $k_r$ represents the scaling factor.

Therefore, the optimization objective function can be transformed as:

$$\min_{U(k)} J = \min_{U(k)} \sum_{i=0}^{N_p-1} \omega_1 \left( v - v_{ref} \right)^2 + \omega_2 \left( SoC - SoC_{ref} \right)^2 + \omega_3 P_g^2 + \omega_4 \Delta F_m^2 + \omega_5 \Delta P_g^2$$

s.t.

$$x_{\min} \le x(k) \le x_{\max}, k = 0, 1, ... N_p - 1$$
$$u_{\min} \le u(k) \le u_{\max}, k = 0, 1, ... N_p - 1$$

(38)

In Equation (38), the first three weights indicate the coordinated relationship among vehicle speed tracking, SoC maintenance, and fuel saving. The last two weights indicate the penalty intensity of driving force variation and engine power variation. Through the cooperation of five weights, it can be realized that the demand power variation can match the engine dynamic characteristics under different transient conditions on the premise of meeting the normal performance requirements.

According to the classification results from Section 3.3, different weight combinations are adopted for different categories to improve the adaptability of the control strategy. The designed weights are shown in Table 4. Working condition categories 1 and 2 represent the low power demand and parked situations, so the weight coefficients are the same, and category 4 represents a medium power demand. The battery discharging capacity is likely to be larger for this condition than for category 1, so the weight $\omega_2$ is increased to reduce SoC fluctuations. Additionally, changes in the driving forces of the driving motors would have a larger impact on the demand power, so $\omega_4$ is increased to raise the penalty for changes in the driving force. Category 3 represents the high-power-demand condition. Changes in the power demand are more severe in this condition, so the vehicle speed tracking penalty is reduced to ensure smooth operation of the power system. In addition, $\omega_2$ and $\omega_4$ are increased to reduce the SoC and power demand fluctuations.

**Table 4.** Weights.

| Classification | $\omega_1$ | $\omega_2$ | $\omega_3$ | $\omega_4$ | $\omega_5$ |
|:---:|:---:|:---:|:---:|:---:|:---:|
| 1 | 14 | 10 | 2 | 9 | 4 |
| 2 | 14 | 10 | 2 | 9 | 4 |
| 3 | 13 | 17 | 2 | 11 | 4 |
| 4 | 14 | 15 | 2 | 10 | 4 |

The co-optimization sampling time is selected to be 0.1 s, the prediction horizon is 10 steps, and the control horizon is five steps.

## 4. Results and Analysis

This section compares the proposed co-optimization strategy with sequential optimization [28] and filter-based control strategy [22,23] in actual working conditions for the heavy-duty off-road vehicle. The superiority of the co-optimization scheme over the other two strategies is clearly demonstrated. The simulation conditions were given in Section 2.5.1. In addition, the hardware used was an Intel(R) Core (TM) i7-10700 CPU @ 2.90 GHz with 33 GB RAM.

### 4.1. Simulation Results

Figure 17 presents the trajectories of vehicle speed, engine power, engine speed, SoC, battery current, and bus voltage using different strategies under the given driving cycle. Figure 17a shows that all control strategies can track the reference vehicle speed well, which demonstrates that the driver's intention is well recognized and fulfilled. Figure 17b,c show that the co-optimization can significantly reduce the transient variation, which reflects the effect of co-optimization of driving force variation and engine power variation through $\omega4$ and $\omega5$. The standard deviation reflects the degree of dispersion of a data set, and the equation for it is shown in Equation (39). As shown in Table 5, the standard deviation of engine speed is used to indicate engine transient variation, and the standard deviations for

the co-optimization decrease by 16.2% and 38.7%, respectively, from that of the sequential optimization and filter-based control strategy.

$$\sigma = \sqrt{\frac{\sum_{i=1}^{n}(x_i - \mu)^2}{n}} \tag{39}$$

where $\sigma$ is the standard deviation, $n$ is the amount of data, $\mu$ is the mean value, and $x_i$ denotes the $i$-th data in data set $x$.

**Table 5.** Comparison of three control strategies.

| | Engine Speed Standard Deviation | Lowest Fuel Consumption Zone Probability (%) | Fuel Consumption (L) | Final SoC (%) | Equivalent Fuel Consumption (L) |
|---|---|---|---|---|---|
| Filter-based control | 223.3509 | 42 | 21.06 | 75.80 | 21.058 |
| Sequential optimization | 163.3545 | 53 | 19.96 | 75.05 | 19.959 |
| Co-optimization | 136.9377 | 55 | 19.33 | 75.12 | 19.327 |

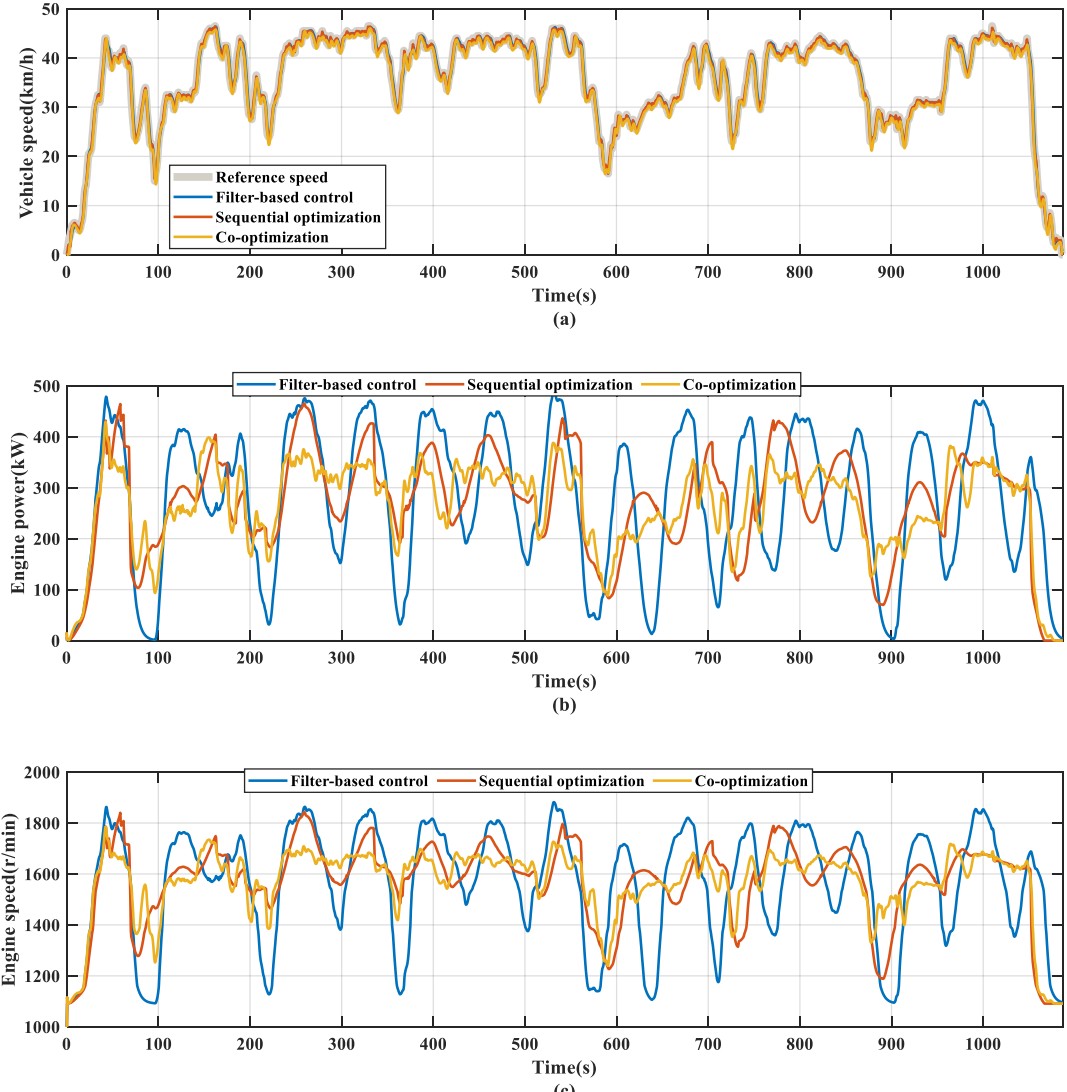

**Figure 17.** *Cont.*

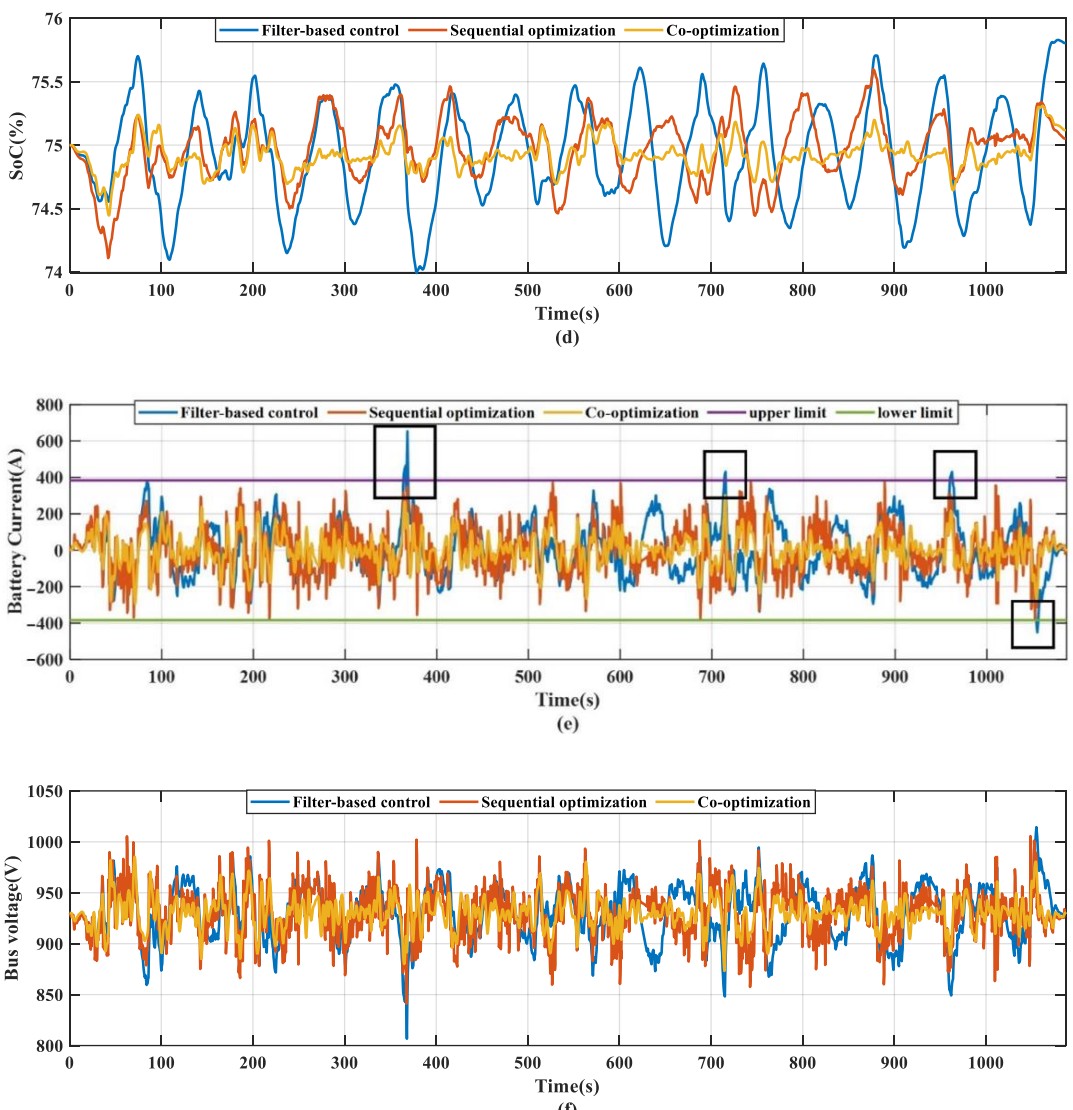

**Figure 17.** Comparison of the three strategies: (**a**) vehicle speed; (**b**) SoC trajectory; (**c**) engine power; (**d**) engine speed; (**e**) battery current (positive represents discharging; negative represents charging); (**f**) bus voltage.

The SoC trajectories are shown in Figure 17d. All three control strategies maintain the SoC at approximately 75%, which is one of the energy management targets for heavy-duty off-road vehicles. However, among three strategies, a narrower range is shown for co-optimization SoC trajectory. Figure 17e,f demonstrate that the co-optimization can realize more mild electrical system operation. As indicated by the black box in Figure 17e, the filter-based control strategy fails to guarantee the battery C-rate within the limits (4C), but the sequential optimization and the co-optimization succeeded.

As is shown in Figure 18, because the co-optimization results in the smallest engine transient variation, its operating points are most closely distributed around the optimal BSFC line. The orange box showed the lowest fuel consumption area. As shown in Table 5, compared to the other two control strategies, the probability of the engine operating points falling within the lowest fuel consumption area increases by 2% and 13% for the co-optimization, respectively.

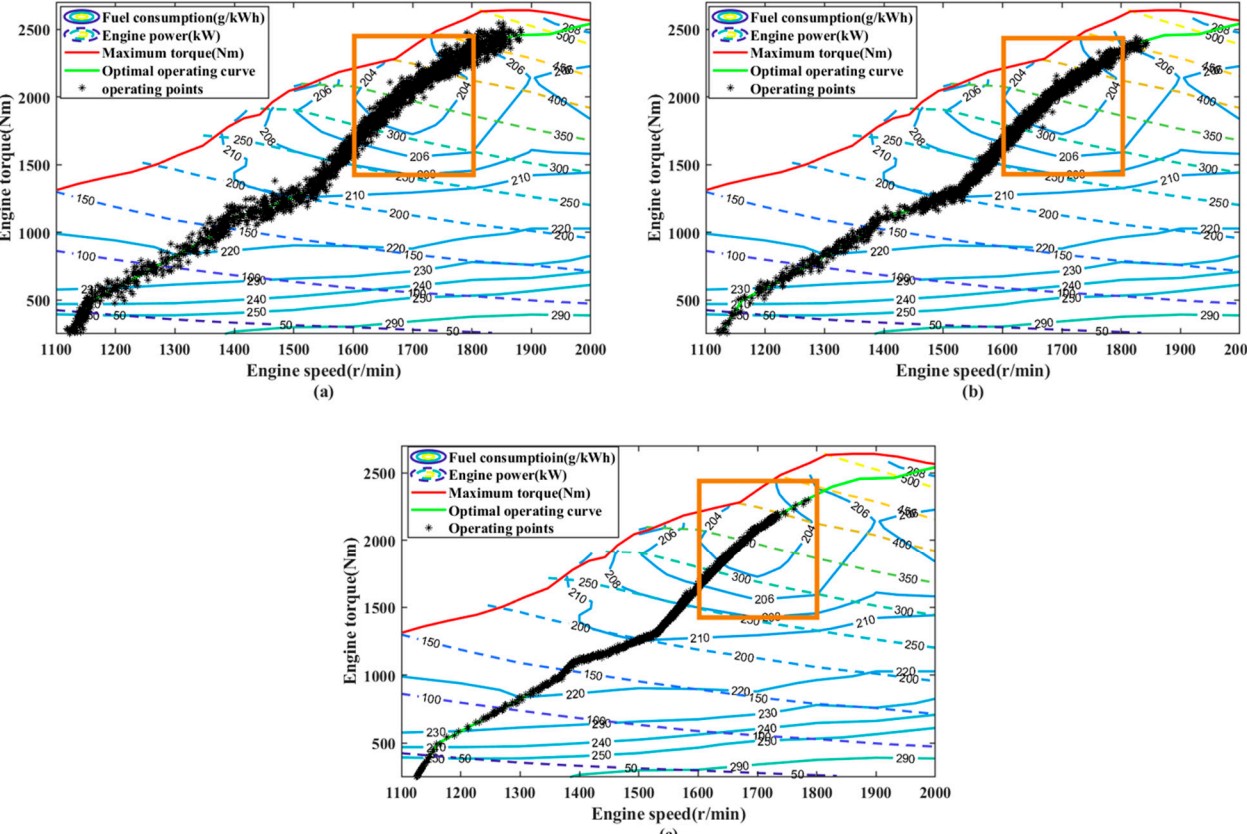

**Figure 18.** Engine operating points: (**a**) filter-based control; (**b**) sequential optimization; (**c**) co-optimization.

In order to clearly demonstrate the effect of the proposed strategy, the partial enlarged view of vehicle speed, motor torque, engine power, and battery current under the conditions of rapid acceleration and deceleration are shown in Figure 19. Figure 19a is the condition of rapid acceleration after deceleration. Because of the phase delay, when the filter-based control strategy is adopted, engine power cannot quickly follow the demand power, so the battery needs to bear a larger part, which causes the battery discharge current to exceed the limit. Figure 19b shows the rapid deceleration condition, and the charging current exceeds the limit because the engine power fails to turn down quickly. When adopting the co-optimization and sequential optimization strategy, because the driving intention is reflected in the predicted vehicle speed, under the comprehensive action of $\omega_1$, $\omega_3$, and $\omega_5$, co-optimization and sequential optimization can quickly adjust the engine power to keep it consistent with the power demand, which becomes an important reason to avoid the battery current exceeding the limit. At the same time, under the condition of rapid acceleration, co-optimization limits the change rate of motor torque through the penalty term $\omega4$, which can smooth the demand power and can further smooth the engine power and battery current output. In the aspect of vehicle speed tracking, co-optimization can still achieve high tracking accuracy, which shows that the negative effects brought by the penalty terms of driving force in the objective function are acceptable.

An overall comparison of the three control strategies is shown in Table 5. Compared with a filter-based control strategy, the co-optimization-based driver-oriented energy management strategy can yield a 38.7% decrease in engine transient variation and an 8.2% decrease in fuel consumption. Furthermore, compared with a sequential-optimization-based control strategy, a 16.2% decrease in engine transient variation and a 3.2% decrease in fuel consumption is achieved. In terms of battery current regulation, the co-optimization strategy effectively avoids excessive battery current.

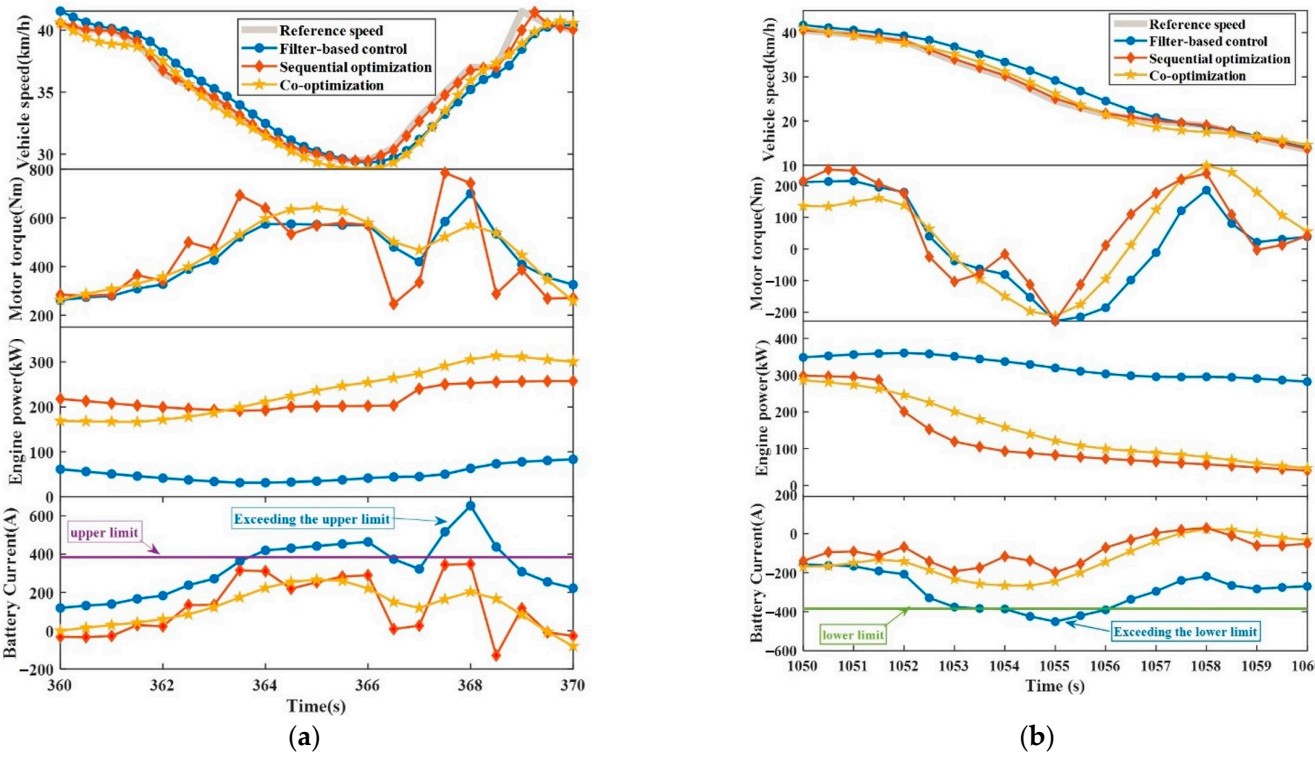

**Figure 19.** Two short intervals: (**a**) accelerating condition; (**b**) decelerating condition.

### 4.2. Hardware-in-the-Loop Test

A hardware-in-the-loop test is conducted to further verify the effectiveness of the proposed strategy. As is shown in Figure 20, the HiL test platform is composed of a real-time target machine, a VCU, a CAN bus, a target machine host PC, and a VCU host PC. The configuration of the components is shown in Table 6.

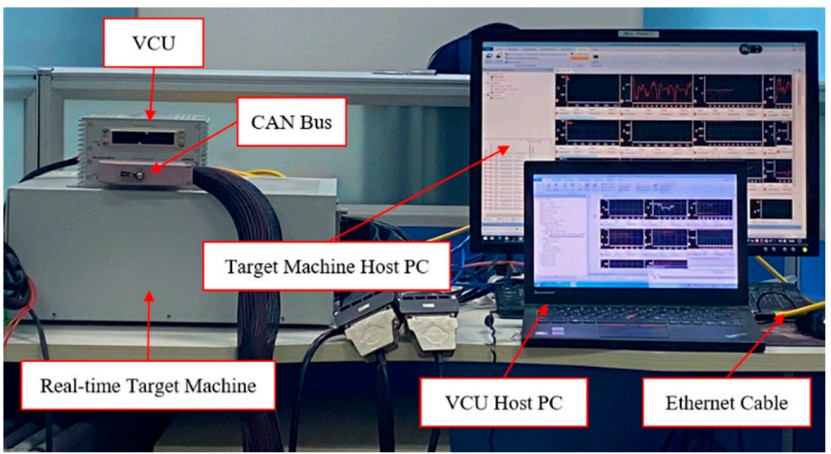

**Figure 20.** HiL test platform.

**Table 6.** The configuration of the HiL platform.

| Component | Configuration |
| --- | --- |
| Real-time target machine | DSPACE PX10 simulation platform |
| VCU | DSPACE MicroAutoBox II |
| Target machine host PC | Intel(R) Core(TM) i7-10700 CPU @ 2.90 GHz 32 GB RAM |
| VCU host PC | Intel(R) Core(TM) i5-5200U CPU @ 2.20 GHz 8 GB RAM |

The co-optimization-based driver-oriented energy management strategy is compiled via automatic code generation technology, and the codes are loaded into DSPACE MicroAutoBox II, which served as the VCU, from the VCU host PC through an Ethernet cable. The plant model and the working conditions' information is compiled and loaded into DSPACE PX10. The DSPACE PX10 and MicroAutoBox II are connected via the CAN bus.

Similar to the way the offline simulation conditions are obtained, the working conditions for the HiL are obtained from actual data, as shown in Figure 6. Moreover, the MPC-based co-optimization control strategy is tested in a real-time environment. Figure 21 shows the performance of the co-optimization control strategy. As shown in Figure 21a, the vehicle can track the reference speed even under drastically changing working conditions. Moreover, the engine works more mildly, effectively avoiding the difficulty in regulating the engine speed, as shown in Figure 21b,c. This means that the energy management strategy is strictly executed such that the SoC can stay stable, as shown in Figure 21d. As a result of perfect engine power tracking, the battery did not have to compensate (or absorb) the lacking (or excessive) power supply. Consequently, the battery current and bus voltage do not exceed the limits, as shown in Figure 21e,f. These results can justify the real-time effectiveness of the proposed strategy.

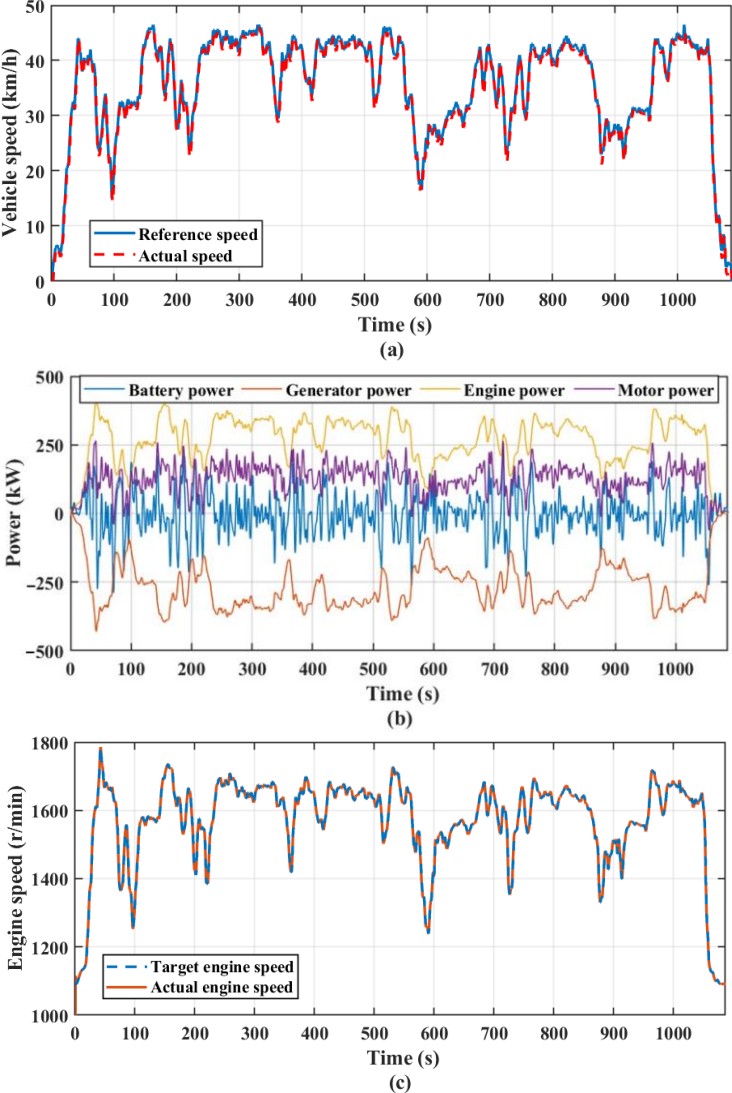

**Figure 21.** *Cont.*

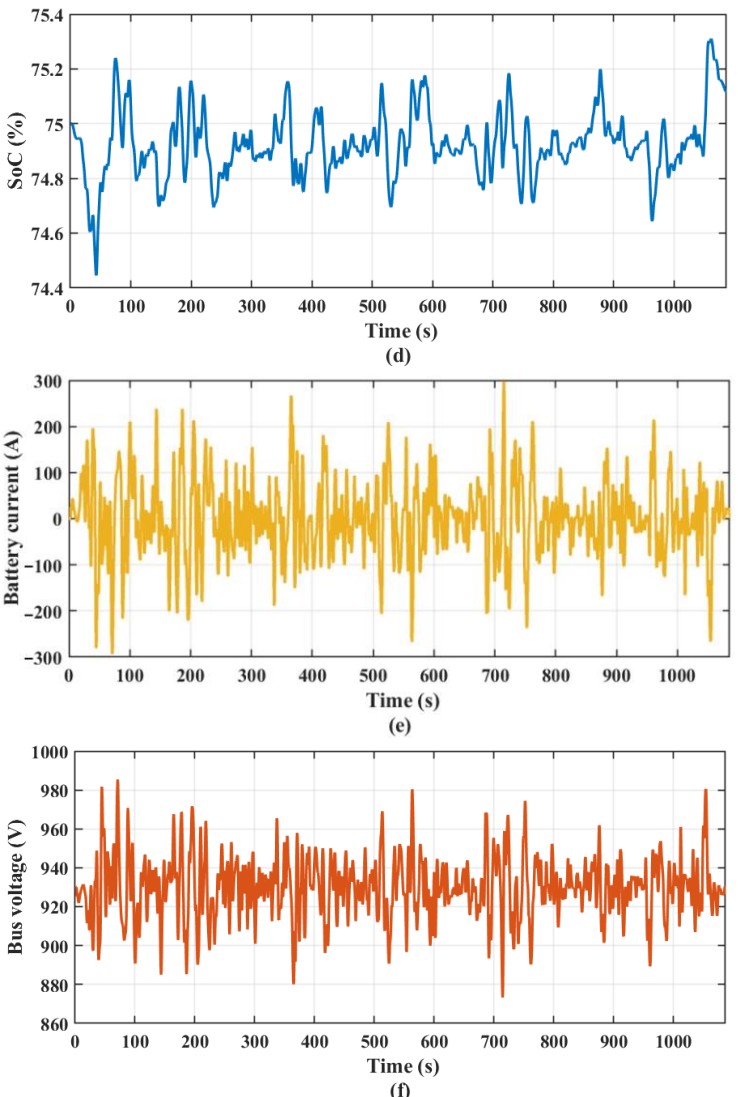

**Figure 21.** HiL test results. (**a**) Vehicle speed; (**b**) power; (**c**) engine speed; (**d**) SoC; (**e**) battery current; (**f**) bus voltage.

## 5. Conclusions

This paper proposed a co-optimization-based driver-oriented energy management strategy to simultaneously ensure engine smooth operation, fuel economy, and battery current regulation for a manned hybrid heavy-duty off-road vehicle operating under aggressive transients. Initially, dynamic models that can accurately reflect the powertrain dynamic characteristics are developed. To recognize the driving intention, a NARX neural network is developed to translate the driving intention into the expected subsequent vehicle speed. An MPC-applied co-optimization-based driver-oriented energy management strategy is designed to achieve vehicle speed tracking and energy management. The NARX helps MPC to control the engine power to follow the demand power. Penalties for driving force variation and engine power variation are added to the objective function to further smooth the demand power. To explicitly consider the influence of road slope in the MPC, a road slope estimator based on a Luenberger observer is developed. K-means clustering is used to classify the vehicle working conditions, and the weights of the co-optimization controller are selected according to the classification results to ensure that the demand power variations can match the engine dynamic characteristic under different transient conditions. The simulation results show that the proposed strategy can satisfy driving intention and maintain SoC at the desired level. Compared with filter-based control

strategy, the engine transient variation is reduced by 38.7%, fuel economy is improved by 8.2%, and the battery overload is effectively avoided. Compared with sequential optimization, the engine transient variation is reduced by 16.2%, and the fuel consumption is reduced by 3.2%. In addition, the HiL test results justify the real-time effectiveness of the proposed strategy.

**Author Contributions:** Conceptualization, X.W. and Y.H.; methodology, X.W.; software, X.W.; validation, X.W.; formal analysis, X.W. and Y.H.; investigation, X.W. and J.W.; resources, X.W.; data curation, X.W.; writing—original draft preparation, X.W.; writing—review and editing, X.W., Y.H. and J.W.; visualization, X.W.; supervision, Y.H.; project administration, X.W.; funding acquisition, Y.H. All authors have read and agreed to the published version of the manuscript.

**Funding:** This research was funded by the National Natural Science Foundation of China, grant number 51475043 and 50975026.

**Institutional Review Board Statement:** Not applicable.

**Informed Consent Statement:** Not applicable.

**Data Availability Statement:** Data will be available on request.

**Conflicts of Interest:** The authors declare no conflict of interest.

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
