# Peer review of "Study on Driver-Oriented Energy Management Strategy for Hybrid Heavy-Duty Off-Road Vehicles under Aggressive Transient Operating Condition"

_sustainability, doi:10.3390/su15097539_

Round 1

Reviewer 1 Report

This paper presents a model predictive controller to balance the engine transients and battery current in heavy-duty off-road vehicles. Overall, this paper presents some new results and insights on hybrid electric vehicles, and the topic discussed is in the scope of Sustainability. My comments are listed below:

(1) The battery current should be suppressed to prolong its lifetime, which is clear. But why do we suppress the engine transient?

(2) It is unclear how the reference speed is obtained.

(3) The HiL setup does not contain any of the real powertrain hardware (engine, motor, battery). What does the HiL test tell us that simulations cannot?

(4) If possible, provide some experiments to show us the effectiveness of the strategy.

(5) Figure 10 is not clear enough, and prediction error may be expressed in a simpler way.

Reviewer 2 Report

This paper presents a comprehensive study of energy management of hybrid vehicles. It contains some very good discussions, including powertrain modeling, co-optimization, and estimation results. There are a few suggested changes to improve this paper.

(1)   Some careful edits are required to avoid typos.

(2) In simulations, how the reference vehicle speed obtained is not elaborated, making readers confused about that.

(3) Why is it necessary to estimate road slope?

(4) What is the fundamental mechanism to save energy based on this control scheme?

(5) For some models, such as the motor model, is that necessary to include all details? Please justify why the studied problem requires a high-fidelity model.

(6) The author needs to explain the meaning of the black box in Figure 16(e), and Table 5 needs to add the results of equivalent fuel consumption.

(7) Page 19, the parameter " Engine speed standard deviation "used to evaluate the engine transient variation should be further explained.

(8) For the energy management of hybrid vehicles, the author can refer to: 1) doi.org/10.1186/s10033-022-00790-5; 2) DOI: 10.1109/TTE.2022. 3141780.

Reviewer 3 Report

Dear authors, please address the following issues and questions and submit the revised manuscript for reconsidering for publication. 

1. This manuscript is about "heavy-duty off-road vehicle", but definition of "heavy-duty off-road vehicle" is not provided in the manuscript. How is "heavy-duty" and "off-road" defined?

2. More detailed information regarding the vehicle model should be provided, such as dimensions (length, width, height), frontal area, drag coefficient of the vehicle.

3. As a series hybrid vehicle, it should have regenerative braking as a standard feature. Authors did not provide detailed information on how the regenerative braking is modeled. 

4. More detailed information of the battery model should be provided. As one of the most important relationships for battery modeling, "OCV vs. SOC" relationship should be presented in the manuscript. Also, is the operating temperature of the battery taken into account? How does the internal resistance of the battery change with temperature?

5. What is the layout in the battery pack of the vehicle model? How many battery cells are series-connected in each string and how many strings are connected in parallel to form the battery pack? The parameters of the battery cell should be provided, instead of only providing the parameters of the entire battery pack. Battery modeling is usually applied to individual battery cell, not the entire battery pack as a whole.

6. Regarding to model validation, how is the "a real vehicle test data set" in Line 295 obtained? From what type of real vehicle? Is it comparable with the simulation data from this manuscript? 

7. Table titles in Line 127 and Line 341 are wrong. Please correct them.

Round 2

Reviewer 3 Report

Authors have carefully addressed all the issues and questions from my last review report. The manuscript has been significantly improved. I recommend this manuscript in current form for publication.